# NATURE: Natural Auxiliary Text Utterances for Realistic Spoken Language Evaluation

**David Alfonso-Hermelo**[*]
david.alfonso.hermelo@huawei.com

**Ahmad Rashid**[*†‡]
ahmad.rashid@huawei.com

**Abbas Ghaddar**[*]
abbas.ghaddar@huawei.com

**Philippe Langlais**[§]
felipe@iro.umontreal.ca

**Mehdi Rezagholizadeh**[*]
mehdi.rezagholizadeh@huawei.com

[*]Huawei Noah's Ark Lab  [†]Vector Institute  [‡]University of Waterloo  [§]RALI/DIRO, Université de Montréal

## Abstract

Slot-filling and intent detection are the backbone of conversational agents such as voice assistants, and are active areas of research. Even though state-of-the-art techniques on publicly available benchmarks show impressive performance, their ability to generalize to realistic scenarios is yet to be demonstrated. In this work, we present NATURE, a set of simple spoken-language oriented transformations, applied to the evaluation set of datasets, to introduce human spoken language variations while preserving the semantics of an utterance. We apply NATURE to common slot-filling and intent detection benchmarks and demonstrate that simple perturbations from the standard evaluation set by NATURE can deteriorate model performance significantly. Through our experiments we demonstrate that when NATURE operators are applied to evaluation set of popular benchmarks the model accuracy can drop by up to 40%.

## 1  Introduction

The past decade has seen a proliferation of voice assistants (VAs) and conversational agents in our daily lives. This has been possible due to the progress in the fields of natural language understanding (NLU), spoken language understanding (SLU) and natural language processing (NLP). Commercial VAs are typically pipeline systems with an NLU engine which attempts to categorize and understand user intent. The main component of the NLU engine is a slot-filling (SF) and intent detection (ID) model. State-of-the-art models [40, 49, 50] generally report a high accuracy and F1 score on popular benchmarks such as ATIS [19] or SNIPS [5] which may give an impression that the problem is solved. However, these benchmarks do not model the distinctive variations of spoken-language and the characteristics that a VA must handle in real scenarios.

It has been observed across the fields of NLP and NLU that state-of-the-art deep learning models fit on the spurious, surface-level patterns of the datasets [33, 55, 22]. A growing body of work has demonstrated this on challenging evaluation sets designed by perturbing an existing evaluation set. These perturbations include character-level additions, deletions and swaps for machine translation [2], character-level adversarial perturbations to trick neural classifiers [9], using a masked language model to generate adversarial perturbations [11] and a heuristic based word substitution model to generate semantically plausible adversarial text [23] among others.

35th Conference on Neural Information Processing Systems (NeurIPS 2021) Track on Datasets and Benchmarks.

| Utterance | Task: | Model Prediction Errors |
|---|---|---|
| play party anthems
→ **ploy** party anthems | ID: | `Play_Music`
→ `Search_Creative_Work` |
| play some sixties music
→ **plays** some sixties music | SF: | [sixties]:`year`
→ [sixties]:`year`; [plays]:`album` |
| listen to dragon ball: music collection
→ **like** listen to dragon ball: music collection | ID: | `Search_Creative_Work`
→ `Play_Music` |
| | SF: | [dragon ball: music collection]:`object_name`
→ [dragon ball]:`artist`; [collection]:`album` |

Table 1: Examples of NATURE-perturbed utterances with badly predicted slots and/or intent. The perturbed utterance is preceded by a →.

Making a large, diverse, spoken-language oriented, multilingual benchmark would be ideal. However this is a labor-intensive, time-consuming, and expensive commitment. As a trade-off between less costly annotation and more realistic data, we propose a framework that focuses on perturbing the existing evaluation set by applying simple, spoken-language oriented, realistic operators[1] that modify the input sentence without perturbing the original meaning. In this paper, we introduce the NATURE (*Natural Alterations of Textual Utterances for Realistic Evaluation*) framework, a compilation of operators that preserve the original semantics while adding realistic spoken-language characteristics to the evaluation set. In addition to producing realistic data, the score analysis of the evaluation sets perturbed with a single operator helps pinpoint the superficial and heuristic dependencies of each model. To the best of our knowledge, no work has attempted to demonstrate that the benchmarks and models for the dual tasks of SF and ID rely on frequent heuristic patterns. Table 1 shows examples of perturbed utterances where a state-of-the-art model [40] correctly predicted the label for the original utterance but failed for the perturbed utterance.

## 2 Related Work

### 2.1 Realizing models use shortcuts

A growing number of studies identify a tendency in NLU models to leverage the superficial features and language artifacts instead of generalizing over the semantic content. A naive way to force generalization is to automatically add noise to the training set, however, as demonstrated by [2], models trained on synthetic noise do not necessarily perform well on natural noise, requiring a more elaborate approach. Given our incapacity to control the features these models learn, each task requires an in-depth analysis and a data or model modification that guides it to the correct answer. For the political claims detection task [38] and [7] unveil a strong bias towards the claims made by frequent actors that require masking the actor and its pronouns during training to improve the performance. Other works have focused on the artifact and heuristic over-fitting for the natural language inference (NLI) task [17, 39, 53, 33, 37] or for the question-answering (QA) task [21]. The work of [12] focuses on the artifacts in named entity recognition task and [1] shows that substituting Named Entities (NEs) influences the robustness of BERT-based models for different tasks (NLI, co-reference resolution and grammar error correction).

### 2.2 Alternative evaluation

Some researchers have proposed evaluation sets with naturally occurring adverse sentences for different tasks such as HANS for natural language inference (MNLI) [33] or PAWS [55] and PAWS-X [51] for paraphrase identification. Another strategy involves a systematic perturbation of the evaluation set [29]. This has gained popularity in recent years with a growing interest in more challenging and adversarial evaluation frameworks. However, a more challenging evaluation set has

---

[1]By realistic, we mean that modified utterances remain semantically similar to the original intention in a real-life scenario.

to ensure high quality annotation, which is why many papers have suggested a human-in-the-loop approach [24, 10, 25]. But these approaches are costly, specially due to the number and quality of annotators necessary to produce a high-quality output. Generalization is more easily achieved when the training data is large and diverse. A model can be effective, yet, if it is only fed with small and/or similar data, it will have difficulties to achieve robustness. Some researchers [32, 54, 6, 35, 36] use data augmentation strategies to improve the training data and help boost a model's performance.

Other researchers follow a different path and suggest evaluating by perturbing the evaluation set using multiple task-agnostic rule-based transformations. These slightly alter the form of the data while affecting very little the semantic content. In this category, we can cite the works of [44] (Checklist tool) and [14] (RobustnessGym).

## 2.3 Spoken-Language perturbation methods

There have been a few works that have done research on spoken-language oriented perturbation methods. Some seek to simulate automatic speech recognition (ASR) errors [48, 46, 28, 16]. Whether using mappings of common ASR errors or based on the acoustic word embedding approach [3], these strategies cannot work for SF and ID because we may loose the token-by-token semantic labeling that is required for SF.

Other works have devised methods that change the sentence form while keeping track of the semantic labeling [52, 27]; although they are not presented as spoken-language oriented. Such approaches, whether they emulate non-native speaker errors or produce counterfactual versions of the original utterances, use value-substitution techniques that require high-quality label-token dictionaries for each new dataset.

In NATURE, we aim to produce spoken-language oriented perturbations [3, 48, 46, 28, 16], such that the utterances remain semantically similar [44, 14]. without using costly label-token dictionaries [34, 30, 29] and human-in-the-loop techniques [24, 10, 25].

## 3  Methodology

We divide the NATURE operators into three categories - fillers, synonyms and *speako* (or similar sounding). Since these operators are intended to introduce human speech inspired small perturbations in SF and ID evaluation, it is desirable for a trained model to maintain its performance under NATURE perturbations. Table 2 gives a few examples of these operators.

### 3.1  Fillers

Fillers are ubiquitous in everyday spoken language and often appear in transcribed human-to-human dialog corpora (such as the Switchboard corpus [13], composed of approximately 1.6% fillers [45]).

Fillers serve as hesitation markers (e.g.: *Bring me the,* ***like,*** *Greek yogurt. I've heard it's really,* ***you know,*** *savoury.*) or as introduction/closure of a turn of speech (e.g., ***Now,*** *bring me the Greek yogurt* ***please and thank you***. ***Actually,*** *I've heard it's really savoury,* ***right?***).

Because they are semantically poor (lacking essential meaning) and therefore do not change the overall meaning of an utterance, fillers are intentionally cleaned off in SF and ID benchmarks. Although we could design a pre-processing step to remove fillers from a VA system it is more interesting to study the impact of fillers and to test the capacity of models, specially those pre-trained on language modeling, to generalize over utterances with fillers.

We propose 4 different filler operators:

- **Beginning-of-sentence** (BOS): a small introductory filler phrase at the beginning of the utterance, such as: *so*, *like*, *actually*, *okay so*, *so okay*, *so basically*, *now* or *well*.

- **End-of-sentence** (EOS): a small conclusive filler phrase at the end of the utterance, such as: *if you please*, *please and thank you*, *if you can*, *right now*, *right away*, *would you mind ?*

- **Pre-verb**: a filler word or sequence of words appearing before the utterance's verb or verbal phrase, such as: *like*, *basically* or *actually*.

| evaluation set | Example sentence |
|---|---|
| Original | add _tune_ to _sxsw fresh_ playlist |
| BOS Filler | **okay so** add _tune_ to _sxsw fresh_ playlist |
| Pre-V. Filler | **like** add _tune_ to _sxsw fresh_ playlist |
| Post-V. Filler | add _tune_ **actually** to _sxsw fresh_ playlist |
| EOS Filler | add _tune_ to _sxsw fresh_ playlist **if you can** |
| Synonym V. | **play** _tune_ to _sxsw fresh_ playlist |
| Synonym Adj. | add _tune_ to _sxsw **cool**_ playlist |
| Synonym Adv. | add **prior** to _sxsw fresh_ playlist |
| Synonym Any | **mix** _tune_ to _sxsw fresh_ playlist |
| Synonym StopW | add _tune_ **the** _sxsw fresh_ playlist |
| Speako | add _**tua**_ to _sxsw fresh_ playlist |

Table 2: Processed variants of original utterances from the SNIPS corpus. The tokens labeled as *music_item* appear with a dotted underline and the tokens labeled as *playlist* show a dashed underline. In SNIPS, the *sxsw* token is part of a playlist name and an abbreviation of *South by Southwest*.

- **Post-verb**: a filler word or sequence of words appearing after the utterance's verb or verbal phrase, such as: *basically*, *actually*, *like* or *you know*.

BOS and EOS operators simply add a filler at the very beginning or the end of the utterance, respectively. The pre-verb and post-verb operators require us to find the part-of-speech (POS) tag of the utterance tokens [2]. Then, the filler is put at the correct place. We add a fail-safe rule to ensure that a filler is added if no verb is found where expected. To that end, we use the overly-recurrent filler, *like*, and the first appearing NE as a pivot instead of the first appearing verb e.g., *let's check **like** avengers)*.

## 3.2 Synonyms

A synonym is a word that can be interchanged with another word in the context, without changing the meaning of the whole. To replicate this semantic operation, we select the POS corresponding to the NATURE operator (verb, adjective, adverb, etc.). Then, we select a word of that type in the input utterance and make a list of corresponding potential synonym candidates (with the same POS tag) to replace it. Next, we use a pre-trained BERT-base model with a language modeling head to produce corresponding probabilities of synonym candidates. We use this BERT-based model instead of a human populated dictionary (such as Wiktionary) since not all dictionary entries show synonyms.

To summarize, we first randomly choose a POS tag and find a target token which has this tag in our utterance. Then we replace the target with a special [MASK] token. We feed this utterance into BERT and obtain a list of candidates with their probabilities.

In case a sentence contains no token with the target POS, we use the more common *noun* POS. We observe an example in the *synonym adv.* row in Table 2.

As we can see in Table 3, not all BERT candidates are suitable synonyms of the target token. We remove candidates that do not have the same POS of the target token. For a better performance, we place each candidate in the sentence before getting its POS. We have 5 different synonym operators based on different target POS: **verb**, **adjective**, **adverb**, **any** (at random between verb, adjective, adverb or noun), **stop-words** (grammatical and most common words).

## 3.3 Speako

Some words sound similar to others but have a different meaning altogether (e.g., *decent* and *descent*, *this* and *these*). This operator is based on the idea that anyone can make an error, but an efficient and robust model should be able to recover a minor mistake using the context. Thus, we introduce speakos (slip of the tongue, speech-to-text misinterpretation). These slips of the tongue appear commonly in

---

[2]We use the NLTK library to find the POS of the tokens.

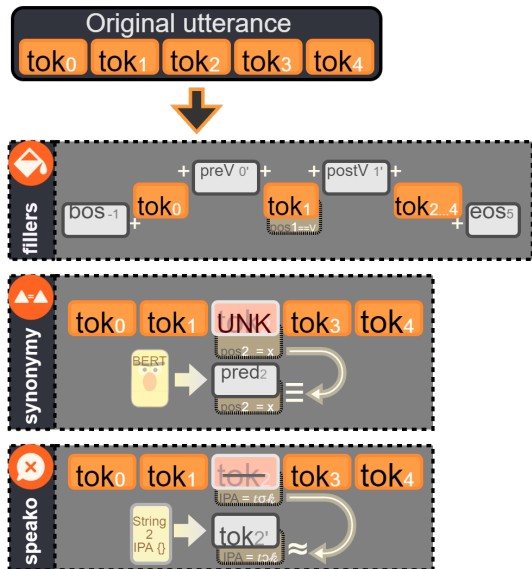

Figure 1: Overview of how each operator alters the original sentence, according to its type (filler, synonymy, speako).

| Token in context | Wiktionary synonyms | BERT candidates |
|---|---|---|
| let me buy it
verb | purchase, accept, [...] | get, buy, present, make, purchase, offer, give, sell, [...] |
| is it large ?
adj | giant, big, huge, [...] | unusual, big, dangerous, large, powerful, [...] |
| i said it quickly
adv | rapidly, fast | fast, well, strong, high, good, deep, large, slow, [...] |
| give me freedom
noun | liberty, license, [...] | rights, property, freedom, status, goods, liberty, [...] |
| i found the ball
stopword | le | the, second, also, third, their, still, a, our, 2nd, [...] |

Table 3: Target words (underlined) of various POS and their synonyms taken from the crowd-sourced dictionary Wiktionary and candidates obtained using a pre-trained BERT language model.

oral human-to-human communication. According to some studies [20, 47] they represent between 48 and 67.4% of all oral errors, depending on the type of speaker. Although we do not find similar studies for user-machine communication, we know this phenomenon is not exclusive to human-to human communication and we expect them to appear in a similar amount.

To implement the speako operator, we use a prepared dictionary of tokens appearing 1000+ times in the whole English Wikipedia[3]. We convert each entry of the dictionary into its representation in International Phonetic Alphabet (IPA). We randomly select one token from the sentence, and also convert it to IPA. We then calculate the similarity between it and the dictionary's entries (using Levenshtein distance) and replace it with the closest candidate. For instance, the sentence *let me watch* (/wɑtʃ/) *a comedy video* could be transformed into *let me which* (/wɪtʃ/) *a comedy video*).

Figure 1 shows how we alter the original utterance by the filler, synonymy and speako operators.

---

[3]We empirically observed that removing all tokens that had a co-occurrence lower than 1000 eliminated most of the nonsensical strings and extreme misspellings and conserved most functional words.

# 4 Experimental Setup

## 4.1 Data

In our work, we use 3 popular open-source benchmarks [4] which are summarized in Table 4:

**Airline Travel Information System (ATIS)** [5] [19] introduced an NLU benchmark for the SF and ID tasks with 18 different intent labels, 127 slot labels and a vocabulary of 939 tokens. It contains annotated utterances corresponding to flight reservations, spoken dialogues and requests.

**SNIPS** [6] [5] proposed the SNIPS voice platform, from which a dataset of queries for the SF and ID tasks with 7 intent labels, 72 slot labels and a vocabulary of 12k tokens were extracted.

**NLU-ED** [7] is a dataset of 25K human annotated utterances using the Amazon Mechanical Turk service [31]. This NLU benchmark for the SF and ID tasks is comprised of 69 intent labels, 108 slot labels and a vocabulary of 7.9k tokens.

Following the common practice in the field [18, 15, 40, 43, 26]), we report the performance of SF using the F1 score. Moreover, we propose an end-to-end accuracy (E2E) metric (sometimes referred in the literature as the sentence-level semantic accuracy [40]). This metric counts true positives when all the predicted labels (intent+slots) match the ground truth labels. This allows us to combine the SF and ID performance in a single more strict metric.

| Benchmark | | Train | Valid. | Eval. |
|---|---|---|---|---|
| ATIS | Sent | 4 478 | 500 | 893 |
| | Words | 50 497 | 5 703 | 9 164 |
| | Voc | 867 | 463 | 448 |
| SNIPS | Sent | 13 084 | 700 | 700 |
| | Words | 117 700 | 6 384 | 6 354 |
| | Voc | 11 418 | 1 571 | 1 624 |
| NLU-ED | Sent | 20 628 | 2 544 | 2 544 |
| | Words | 145 950 | 18 167 | 17 347 |
| | Voc | 7 010 | 2 182 | 2 072 |

Table 4: Dataset size information of ATIS, SNIPS and NLU-ED benchmarks.

Any dialog-based dataset extracted from real user situations has the potential of containing private and security sensitive information. This is the main cause for the relatively low amount of datasets for SF and ID. The benchmarks we mention are well known and cautiously cleaned (as presented in Section 3). NATURE operators purposely avoid using any type of resource that would contain personal information. To the best of our knowledge, our work is not detrimental to people's safety, privacy, security, rights or to the environment in any way.

## 4.2 Models

We use two different state-of-the-art models:

**Stack-Prop+BERT** [40] uses BERT as a token-level encoder that feeds into two different BiLSTMs, one per each task. The output of the SF BiLSTM is added to the ID BiLSTM input in order to produce a token-level intent prediction which is further averaged into a sentence-level prediction.

**Bi-RNN** [49] uses two correlated BiLSTMs that cross-impact each other by accessing the other's hidden states and come to a joint prediction for SF and ID.

---

[4] We do not consider datasets for other VA related tasks such as multi-intent detection (e.g., MixATIS and MixSNIPS [41]) or multi-turn dialog (e.g., SGD dataset [42]).

[5] CGNU General Public License, version 2

[6] Creative Commons Zero v1.0 Universal License

[7] Creative Commons Attribution 4.0 International License

The pre-trained version of these models were not available[8]. For ATIS and SNIPS, we trained the models using the same hyperparameters proposed in the documentation by [40][9] and [49][10], respectively. For NLU-ED, we use the hyperparameters from SNIPS, as their size is comparable. The trained models obtained comparable results to their published counterpart (see in Appendix Section A). To train the models, we used 1 NVIDIA Tesla V100. It took between 3 and 71 hours to train the Stack-Prop+BERT model [40] (depending on the size of the benchmark), and between 68 and 130 hours to train the Bi-RNN model [49].

## 4.3 Modified NATURE Evaluation Sets

Since the original evaluation sets only cover a limited set of patterns, we transform them by applying the NATURE patterns to obtain evaluation sets of the same size as the original ones. As previously illustrated, NATURE operators offer simple ways of perturbing utterances. In order to avoid rendering utterances unrecognizable from their original version, we only apply one operator at a time and only once in the sentence (e.g. we add 1 filler or synonymize 1 token or transform 1 token into its speako version). We design 2 NATURE experimental evaluation sets: *Random* and *Hard*. In the Random setting, for each utterance, we apply one operator at random and we repeat the random operator selection 10 times and calculate the mean score.

For the Hard setting, we use the popular BERT fine-tuning model [8] [11] to filter-in the most challenging operators. For each evaluation utterance, we select the operator with the lowest confidence score (probability of the true class). In Table 5 we show the operator composition (by percentage) of the Hard evaluation sets for each dataset.

The Random evaluation set is meant to show how a random small change in the sentence can influence evaluation while the Hard evaluation set is meant to assess the lower-bound performance of how much the model depends on similar pattern sentences to obtain the correct prediction.

| Operator | ATIS | SNIPS | NLU-ED |
|---|---|---|---|
| BOS Filler | 0.8 | 0.1 | 2.5 |
| Pre-V. Filler | 6.0 | 3.7 | 16.0 |
| Post-V. Filler | 1.9 | 8.6 | 5.1 |
| EOS Filler | 9.0 | 52.3 | 8.3 |
| Syn. V. | 25.6 | 5.4 | 16.3 |
| Syn. Adj. | 29.2 | 15.0 | 23.4 |
| Syn. Adv. | 11.8 | 5.6 | 10.2 |
| Syn. Any | 5.3 | 1.1 | 4.8 |
| Syn. StopW | 3.2 | 2.7 | 6.4 |
| Speako | 7.2 | 5.4 | 6.9 |

Table 5: Composition (by percentage) of JointBERT-selected operators for the Hard experimental evaluation set.

## 5 Results and Discussion

### 5.1 Qualitative Evaluation

Our assumption is that the operator-generated utterances share the same meaning and labeling as the original sentence. In order to measure this, we conducted a small but representative multiple-choice survey. We select 120 operator-perturbed utterances from the ATIS, SNIPS and NLU-ED benchmarks. We selected at random 40 utterances from each benchmark, making sure they were also evenly distributed between operators (12 utterances per operator). In addition to these, we cherry-picked 12 original utterances of high-quality that served as control. As we can see in the Survey Table in the

---

[8]`https://github.com/LeePleased/StackPropagation-SLU` and `https://github.com/ray075hl/Bi-Model-Intent-And-Slot`

[9]300 epochs, 0.001 learning rate, 0.4 dropout rate, 256 encoder hidden dimensions, 1024 attention hidden dimensions, 128 attention output dimensions, 256 word embedding dimensions for ATIS and 32 for SNIPS.

[10]500 epochs, max sentence length of 120, 0.001 learning rate, 0.2 dropout rate, 300 word embedding size, 200 LSTM hidden size

[11]More specifically, JointBERT [4] implemented at `https://github.com/monologg/JointBERT`.

Appendix Section C, the control scores stayed high and therefore, there was no reason to invalidate any participant's annotations.

14 participants (NLP and ML researchers, with no links to this work) volunteered to participate in this unpaid survey and consented verbally to the use of their data within the scope of this research. To avoid a decrease in annotation quality (due to fatigue), we split the participants in 2 groups of 7 members and divided the utterances in two sets (each with 60 operator-perturbed + 12 control utterances). We estimated the survey time to be 30-60 minutes, which was not far from the actual time (27-53 minutes).

For each utterance, we asked the participants to evaluate the intent and slot labels as *reasonable* or *unreasonable*.

|  | Group 1 | | Group 2 | |
|---|---|---|---|---|
|  | Experiment | Control | Experiment | Control |
| Slot | 94.5 | 94.0 | 93.8 | 97.0 |
| Intent | 89.0 | 97.6 | 85.9 | 97.5 |

Table 6: Survey results and statistics per group. All scores appear as percentages and indicate how the samples were perceived. A lower score indicates that more tokens and utterances have an *unreasonable* label.

In Table 6 we observe a sizable decrease on the experiment side for Intent, which can be partially explained by the disposition of some operators to perturb word types (such as verbs) that are highly associated with the intent classification. We also observe that the Slot labeling results are high and very close to the control scores. This indicates that (contrary to many DA strategies) the NATURE operators maintain a close-to-ground-truth slot labeling.

## 5.2 Quantitative Evaluation

Table 7 shows the performances of the Stack-Prop+BERT and Bi-RNN models trained on the original train data of ATIS, SNIPS and NLU-ED benchmarks. Models are evaluated on the Original, Random and Hard evaluation sets. We also show the scores on 10 evaluation sets, each perturbed with a single NATURE operator, where one operator is applied once to each utterance of the evaluation set. In Table 7, for each benchmark, we report the F1 and accuracy on the SF and ID tasks respectively, and the E2E metric. Furthermore, we report the unweighted average (Avg. column) of the aforementioned scores on the three benchmarks. The perturbed evaluation set results are sorted in descending order according to the averaged E2E metric. We notice that Stack-Prop+BERT outperforms Bi-RNN not only on original, but also on all evaluation set variants. More precisely, we observe a gap of 6.3%, 8.7% and 5.9% on the *Avg.* E2E metric on the Orig, Random and Hard evaluation sets.

First, we observe a noticeable lowering in the scores on Random, and quite a radical change on Hard. We consider the possibility that the Hard evaluation set incorporates more noise than the Random evaluation sets, and this could be the cause of this low score. However, depending on the benchmark, the sharpest operators are not always the ones expected to be most disruptive. Yet, the decrease in score is extreme across all benchmarks and for both models.

As mentioned earlier, fillers contribute little to the semantics of an utterance and should not be disruptive for the model. The speako operator is more disruptive semantically, specially for the cases where the original token cannot be deduced from the context and the perturbed token. We expect the synonym operators to be the most disruptive of the three since we modify the semantic value of a whole word at a time. In Table 7 we observe that the model handles most filler operators reasonably well, however, we are surprised to see the scores drop considerably for the EOS. As shown on Table 7, the EOS operator drops the E2E accuracy of both models by about 40% on average across all benchmarks. This suggests some syntax-level pattern dependence where the models use the position of the tokens to achieve the correct predictions. The synonym operators, specially the adverb and adjective, greatly deteriorate the performances. This decrease in score shines a light on the importance of the token-level pattern, signaling that the models are using certain adjectives and adverbs to make their predictions. Since, in the benchmarks, adjectives and adverbs are much less diverse than the nouns and verbs, we infer that the models are using these words as prediction clues.

| Evaluation Set | ATIS | | | SNIPS | | | NLU-ED | | | Avg. | | |
|---|---|---|---|---|---|---|---|---|---|---|---|---|
| | Slot (F1) | Intent (Acc) | E2E (Acc) | Slot (F1) | Intent (Acc) | E2E (Acc) | Slot (F1) | Intent (Acc) | E2E (Acc) | Slot (F1) | Intent (Acc) | E2E (Acc) |
| Stack-Prop+BERT | | | | | | | | | | | | |
| Orig | 95.7 | 96.5 | 86.2 | 95.0 | 98.3 | 87.9 | 74.0 | 85.1 | 67.8 | 88.2 | 93.3 | 80.6 |
| Rand | 91.3 | 95.0 | 66.5 | 83.4 | 96.1 | 53.8 | 67.4 | 76.1 | 56.8 | 80.7 | 89.1 | 59.0 |
| Hard | 82.3 | 90.7 | 34.9 | 70.6 | 95.3 | 12.9 | 55.5 | 62.7 | 38.9 | 69.5 | 82.9 | 28.9 |
| Pre-V. Filler | 95.6 | 96.5 | 85.6 | 92.2 | 98.3 | 79.3 | 71.0 | 83.6 | 65.7 | 86.3 | 92.8 | 76.9 |
| Syn. StopW | 93.0 | 94.8 | 76.5 | 89.7 | 96.7 | 74.3 | 70.2 | 78.9 | 60.2 | 84.3 | 90.1 | 70.3 |
| BOS Filler | 95.6 | 96.2 | 85.8 | 86.5 | 97.1 | 54.9 | 72.5 | 80.8 | 63.9 | 84.9 | 91.4 | 68.2 |
| Post-V. Filler | 94.0 | 96.5 | 80.3 | 84.8 | 98.0 | 57.1 | 68.0 | 84.1 | 63.6 | 82.3 | 92.9 | 67.0 |
| Syn. V. | 90.1 | 95.3 | 63.6 | 88.4 | 95.1 | 66.7 | 68.5 | 74.2 | 56.5 | 82.3 | 88.2 | 62.3 |
| Speako | 92.9 | 92.7 | 72.5 | 77.9 | 94.6 | 45.3 | 69.5 | 74.2 | 57.6 | 80.1 | 87.2 | 58.5 |
| Syn. Any | 90.3 | 90.5 | 54.4 | 86.9 | 94.4 | 61.6 | 67.8 | 71.0 | 53.5 | 81.7 | 85.3 | 56.5 |
| Syn. Adj. | 84.7 | 92.7 | 42.4 | 78.2 | 95.4 | 44.4 | 60.2 | 69.7 | 47.2 | 74.4 | 85.9 | 44.7 |
| Syn. Adv. | 88.2 | 89.1 | 43.9 | 77.6 | 94.3 | 41.9 | 61.6 | 65.6 | 45.4 | 75.8 | 83.0 | 43.7 |
| EOS Filler | 88.9 | 96.3 | 54.1 | 72.1 | 97.7 | 13.1 | 63.9 | 78.0 | 53.6 | 75.0 | 90.7 | 40.3 |
| Bi-RNN | | | | | | | | | | | | |
| Orig | 94.9 | 97.6 | 84.7 | 89.4 | 97.1 | 76.6 | 66.4 | 80.9 | 61.7 | 83.6 | 91.9 | 74.3 |
| Rand | 89.9 | 94.3 | 61.8 | 75.6 | 94.1 | 39.0 | 60.6 | 70.8 | 50.1 | 75.4 | 86.4 | 50.3 |
| Hard | 79.9 | 92.0 | 27.6 | 62.4 | 92.9 | 7.0 | 49.6 | 58.8 | 34.4 | 64.0 | 81.2 | 23.0 |
| Pre-V. Filler | 94.7 | 97.3 | 82.2 | 84.6 | 96.4 | 60.0 | 63.3 | 80.1 | 59.3 | 80.9 | 91.3 | 67.2 |
| Syn. StopW | 90.6 | 94.7 | 72.7 | 80.5 | 95.4 | 56.4 | 62.3 | 73.2 | 52.7 | 77.8 | 87.8 | 60.6 |
| BOS Filler | 80.7 | 96.7 | 82.6 | 80.9 | 96.7 | 38.4 | 65.8 | 78.8 | 59.6 | 75.8 | 90.7 | 60.2 |
| Post-V. Filler | 93.8 | 96.9 | 80.3 | 77.9 | 96.6 | 37.4 | 62.6 | 79.3 | 56.6 | 78.1 | 90.9 | 58.1 |
| Syn. V. | 87.6 | 95.9 | 56.6 | 79.5 | 92.1 | 50.6 | 61.3 | 70.5 | 50.7 | 76.1 | 86.2 | 52.6 |
| Speako | 91.8 | 90.3 | 68.1 | 70.1 | 90.1 | 33.6 | 61.5 | 69.8 | 51.0 | 74.5 | 83.4 | 50.9 |
| Syn. Any | 89.2 | 90.4 | 52.6 | 77.8 | 91.4 | 40.6 | 62.0 | 67.3 | 49.1 | 76.3 | 83.0 | 47.4 |
| Syn. Adj. | 81.7 | 94.2 | 34.4 | 71.7 | 93.9 | 34.9 | 54.3 | 65.5 | 42.1 | 69.2 | 84.5 | 37.1 |
| Syn. Adv. | 87.2 | 85.1 | 38.4 | 69.9 | 92.1 | 29.0 | 54.7 | 61.4 | 40.3 | 70.6 | 79.5 | 35.9 |
| EOS Filler | 88.9 | 96.8 | 52.2 | 64.1 | 94.1 | 5.9 | 56.4 | 65.8 | 42.0 | 69.8 | 85.6 | 33.4 |

Table 7: SF, ID and E2E performances of BERT and RNN based models trained on ATIS, SNIPS, and NLU-ED and evaluated on their original and NATURE-perturbed evaluation sets. We show results on *per-operator* as well as on Random and Hard evaluation sets. Furthermore, we report the unweighted average score on the 3 benchmark we considered. The lowest scores in each column appear underlined.

| Evaluation Set | ATIS | | | SNIPS | | | NLU-ED | | | Avg. | | |
|---|---|---|---|---|---|---|---|---|---|---|---|---|
| | Slot (F1) | Intent (Acc) | E2E (Acc) | Slot (F1) | Intent (Acc) | E2E (Acc) | Slot (F1) | Intent (Acc) | E2E (Acc) | Slot (F1) | Intent (Acc) | E2E (Acc) |
| Stack-Prop+BERT | | | | | | | | | | | | |
| Orig | 95.7 | 96.5 | 86.2 | 95.0 | 98.3 | 87.9 | 74.0 | 85.1 | 67.8 | 88.2 | 93.3 | 80.6 |
| Checklist Contract. | 95.6 | 96.6 | 85.8 | 94.6 | 98.2 | 86.8 | 73.8 | 84.6 | 67.4 | 88.0 | 93.1 | 80.0 |
| Checklist NER | 94.7 | 96.5 | 84.6 | 92.9 | 98.2 | 83.0 | 73.7 | 85.1 | 67.6 | 87.1 | 93.3 | 78.4 |
| Checklist Typo | 85.1 | 92.2 | 51.0 | 78.3 | 95.4 | 51.7 | 57.1 | 70.8 | 46.9 | 73.5 | 86.1 | 49.9 |
| Checklist Punct. | 85.2 | 96.6 | 42.8 | 71.7 | 97.7 | 20.4 | 55.4 | 26.3 | 16.7 | 70.8 | 73.5 | 26.6 |
| Bi-RNN | | | | | | | | | | | | |
| Orig | 94.9 | 97.6 | 84.7 | 89.4 | 97.1 | 76.6 | 66.4 | 80.9 | 61.7 | 83.6 | 91.9 | 74.3 |
| Checklist Contract. | 94.8 | 97.5 | 84.3 | 88.9 | 96.5 | 75.8 | 67.0 | 81.1 | 61.0 | 83.6 | 91.7 | 73.7 |
| Checklist NER | 93.8 | 97.5 | 83.0 | 89.5 | 96.7 | 78.2 | 67.6 | 81.7 | 61.5 | 83.6 | 92.0 | 74.2 |
| Checklist Typo | 81.6 | 92.1 | 43.2 | 70.1 | 92.7 | 37.8 | 49.5 | 66.4 | 41.6 | 67.1 | 83.7 | 40.9 |
| Checklist Punct. | 87.4 | 96.7 | 40.7 | 71.0 | 96.2 | 20.7 | 50.0 | 41.4 | 22.4 | 69.5 | 78.1 | 27.9 |

Table 8: SF, ID and E2E performances of BERT and RNN based models trained on ATIS, SNIPS, and NLU-ED and evaluated on their original and CHECKLIST-perturbed evaluation sets. The lowest scores in each column appear underlined.

The speako operator scores suggest a good capacity of the models to overcome these variants and generalize using the remaining context.

Interestingly, we notice that the drop of performances is highly strong on the E2E metric. For instance, using the Stack-Prop+BERT model on the ATIS evaluation set, perturbed with the `EOS filler` operator, we observe a 0.3% and 6.8% drop on SF and ID respectively but a 32.1% drop on E2E. We use the E2E metric since it is more representative of the whole frame accuracy of real world scenarios [15, 40], where a VA can only execute the expected command if the intent and all slots are correctly predicted. A more concise illustration of Table 7 's results is shown on Figure 2 in the Appendix Section D.

We also apply a general perturbation method, the Checklist tool [44][12] to evaluate model performance on the three benchmarks. Although not designed for spoken language, some of the operators are useful to diagnose the problem of over-fitting to spurious patterns and correlations. We use 4 that are suitable for most sentences: the **punctuation** operator removes or adds a final punctuation according to its presence or absence in the text, the **typo** operator swaps random characters with one of its neighbours, the **contraction** operator replaces contracted words with their non-contracted version or vice-versa (e.g., *don't → do not*, *cannot → can't*), the **NER** operator detects and replaces first names, locations and numbers with other named entities of the same type. Table 8 shows the results and we observe that the punctuation operator can reduce the E2E accuracy of both models by more than 45% on average across all benchmarks. This supports the results we observed on NATURE.

## 6   Conclusions

Neural Network models have a black-box architecture that makes it hard to discern when they correctly generalize over the input and when they resort to heuristic features that correlate to the expected output.

We present the NATURE operators, apply them to evaluation sets of standard SF and ID benchmarks and observe a significant drop of the state-of-the-art model scores. The different operators in our framework help discern what surface patterns the model is exploiting.

These results should hopefully encourage the development of better, more challenging benchmarks and the search for more robust models, capable of handling more realistic, fitting and spoken-language oriented utterances.

For future work, we wish to expand the NATURE operators to include speech impediments (such as lisp, stutter and dysarthria), extend the operators to be multi-lingual and work on multi-turn dialogue and multi-intent detection tasks.

## 7   Acknowledgments

We would like to thank the team at Mindspore[13], a new deep learning computing framework, for partial support on this work. Moreover, we want to thank Prasanna Parthasarathi for his valuable feedback and suggestions and the survey volunteers for their time and participation.

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
