# OpenReview forum: "NATURE: Natural Auxiliary Text Utterances for Realistic Spoken Language Evaluation"
_NeurIPS.cc/2021/Track/Datasets_and_Benchmarks/Round2 — NeurIPS 2021 Datasets and Benchmarks Track (Round 2)_

### Official Review · Reviewer_FSdF · 2021-09-17
**A valid and fair set of operations on language datasets that decreases the performance of the SOTA models on two tasks**

**Rating:** 7
**Confidence:** 4

**Strengths:**

1.  The data augmentation and transformations introduced by this paper are valid and can happen in real-life spoken and even written language.
2.  This paper as a benchmark can be an incentive for researchers to develop more generalizable language models.
3.  The evaluations and analysis are extensive and very well done.
4.  The low score when the EOS operator is used is interesting and can be further investigated.



**Weaknesses:**

1.  The paper is already very good, but it would have been stronger if the authors could use models that have pre-trained versions available to ensure that implementation discrepancies do not affect the comparability of the performances.
2.  Adding a paragraph in the related work section to describe how this paper is different from the cited works could be helpful.

Some minor and easy-to-fix issues:

1.  Although it is mentioned in lines 222-223, it would be better to describe what the percentages are showing in the caption of table 6 so that reader can understand without searching for the description outside the caption.
2.  Table 5 is not referenced in the body of the paper. All figures and tables should be referenced to let the readers know when to look at them.



**Additional Feedback:**

The link you provided in the comments does not work. I do not know whether it is the same material that you included in the GitHub repository or not.

**Clarity:**

The paper is enjoyable and easy to read. The flow of the paper is good and keeps the reader engaged.


**Correctness:**

The evaluations are thorough and valid. The claims are supported by the results.

**Documentation:**

The authors provide a link to the dataset, but they do not include the code.


**Ethics:**

This paper uses publicly available datasets and models, and therefore, it does not introduce any ethical concerns on its own unless it stems from the previous works they used in this paper.


**Relation To Prior Work:**

Authors position their work very well with respect to related works in the area.


**Summary And Contributions:**


This work presents a set of language operators/transformations called NATURE to transform the sentences in the evaluation/test set of different language datasets and evaluates the state-of-the-art models on the perturbed sentences for intent detection (ID) and slot filling (SF) tasks.

NATURE operators are categorized into three groups including *Fillers*, *Synonyms*, and *Speako (similar sounding).*
The authors hypothesize that a trained model should have the same performance when the NATURE operators are applied to the data.

The authors mention that researchers could simply remove fillers as a pre-processing step, but it is more interesting to study their effects.
They divide **fillers** into four categories:

1.  Beginning-of-sentence (BOS)
2.  End-of-sentence (EOS)
3.  Pre-verb
4.  Post-verb

The first two perturbations are easy to add, but the last two perturbations require finding the part-of-speech (POS) tags. They also have a fail-safe option to add the common filler &ldquo;*like*&rdquo; when no verb is found.

For **synonyms**, they randomly choose a POS tag and then find a word from the utterance that has this POS tag, and they replace this token with the [MASK] token. Finally, they feed this utterance to BERT which outputs a list of candidates with their corresponding probabilities. Some of these candidates are not appropriate substitutions. To remedy this, the authors remove candidates that do not have the same POS tag.
If an utterance doesn&rsquo;t contain the randomly chosen POS tag, they use the common *noun* POS tag.

For **speako**, they make a list of words that occur more than 1000 times in the whole English Wikipedia and convert each of them to their International Phonetic Alphabet (IPA) representation. Next, they randomly select one word from the sentence, convert it to its IPA format, and compute the similarity between it and all items in the list using Levenshtein distance. Finally, they replace the word from the utterance with the closest candidate from the list.

They only apply one of the transformations to each utterance to keep perturbed sentences recognizable and comparable to their original formats.
The authors design two different evaluation sets using NATURE operators which they call *Random* and *HARD*. In the Random design, the authors select one operator randomly for each utterance, repeat it ten times, and compute the average score. In the HARD design, they use the JointBERT model to select the most challenging operator, and for each utterance, they select the operator with the lowest confidence score.

The authors assume that applying NATURE operators on the utterances maintains the semantics, and the resulting utterances are similar to original utterances in terms of meanings and labeling. To measure this, they provide a multiple-choice survey. The result of the survey shows high scores in both groups of participants for slot-filling while a lower score for intent detection.

Authors also run another set of experiments with a general perturbation method, *the Checklist tool* which includes four simple perturbations; (1) adding or removing **punctuations**, (2) adding **typo** by swapping random characters with one of their keyboard neighbors, (3) **contraction** operator that swaps contracted words with their non-contracted version and vice versa (4) **NER** operator that substitutes names, location, and numbers with other named entities.

They test their method on three datasets including Airline Travel Information System (ATIS), SNIPS voice platform, and NLU-ED.
They evaluate the performance of slot-filling by reporting the F1 score, and the performance of intent detection by reporting accuracy. They also use an end-to-end accuracy (E2E) which counts all true positives for both slot-filling and intent detection and reports one metric for the two tasks of SF and ID.

The authors use two models; Stack-Prop+BERT and Bi-RNN.
Stack-Prop+BERT uses BERT to encode tokens and then feeds them into two BiLSTMs, one for the SF task and another for the ID task. The output of the SF BiLSTM is added to ID BiLSTM input to be able to generate word-level intent labels which are then averaged to produce a sentence-level label.

The results consistently show a lower performance on the perturbated data compared to the original utterances. This supports the claim that the state-of-the-art models cannot generalize well to small changes in language, and shows the utility of this work.

---

> ### Author Response · Authors · 2021-09-29
> **Issues corrections**
>
> In response to the weakness point 1. it would have been stronger if the authors could use models that have pre-trained versions available to ensure that implementation discrepancies do not affect the comparability of the performances: We agree and appreciate the suggestion. Unfortunately, the state-of-the-art models we use have no pre-trained versions. Although, since the hyper-parameters were specified in the model’s Github pages, we replicated the models as closely as possible.
>
> In response to the weakness point 2. adding a paragraph in the related work section to describe how this paper is different from the cited works could be helpful: using the additional content page allowed, we added a description at the end of the Related Works section highlighting the main similarities and divergences between NATURE and the cited works.
>
> In response to the minor issue 1. it would be better to describe what the percentages are showing in the caption of table 6: we added the small change to Table 6 caption.
>
> In response to the minor issue 2. table 5 is not referenced in the body of the paper: you are right, we have corrected this.
>
> In response to the additional feedback 1. The link allows to download the scripts used to modify and make the dataset. Unfortunately the link we can provide right now (before official release) is time-limited. When we try to add a new comment and update it the system gives us an error. We have notified the Neurips Dataset and Benchmark committee. The new link is :
>
> https://etrans8.huawei.com/valid.aspx?d=mVQ40aG3CGx9ezeQsBj/6VcQ7aEcb/xMVn/gJEQi2CHd0vtdfssOGyVew19JkSP+gcKlYQDxXiKYAJOL1tkdOHZh3XPiUTzI0xf3CQf1B5BaB8WYLb6q2KAE/FVmruxFcTti8l6Ye36hU93ajz9f03Rozt1yn53xxy0RtRDxJiY7koeNIao/1E2JbwbQAoBKXGSWdKkSxNBnT7zvVdOWbg65ZA8emfT98iEQ/uUlQUxddYLENPtTukbf4DjRS47gwdw6q/K5/rDwqCkyf5mpeSWk9MoD7v4xA7WW9efu47Uw7nfYKpz8sfub4IvCW/s4HO6sqoaFSUBePvjN0D+Zbg==

---

### Official Review · Reviewer_gBoB · 2021-09-20
**A set of spoken language transformations that maintain semantic meaning to evaluate and improve generalization**

**Rating:** 6
**Confidence:** 3

**Strengths:**

NATURE provides a way to automatically modify sentences in ways that maintain semantic meaning to add more natural spoken language characteristics.
The paper does show that current models are not capturing the semantic meanings, since changing data in ways that does not change the semantic meaning of the statements can significantly decrease their performance.


**Weaknesses:**

Incorporating the work done showing NATURE’s use as a tool to augment datasets into the main paper would make a stronger argument for its utility.


**Additional Feedback:**

None

**Clarity:**

 The paper was clearly written and easy to follow.


**Correctness:**

 The evaluation methods and experiments were appropriate and performed correctly.


**Documentation:**

The modified dataset used in the analysis is available on GitHub, and the process used to generate it was clearly described.


**Ethics:**

There were not any ethical concerns.


**Relation To Prior Work:**

Previous work has suggested the use of human in the loop approach to add additional adverse data, while this work provides an automatic method.


**Summary And Contributions:**

This paper presents NATURE a set of spoken language transformations to modify a sentence without changing the semantic meaning. Current deep learning models overfit on surface level patterns leading to poor generalization. NATURE is used to modify the data from several existing slot-filling and intention detection benchmarks and is compared to models trained on the original unmodified benchmarks. When evaluated end to end the performance of the models was severely diminished.

---

> ### Author Response · Authors · 2021-09-29
> **Unsure of the weakness point to work on**
>
> In response to the weakness point of incorporating the work done showing NATURE’s use as a tool to augment datasets into the main paper: We are not completely sure we understood the reviewer. We think that the reviewer’s suggestion is to include the appendix section D on Data Augmentation within the main paper. In Section D we show that if we use Data Augmentation during training, we can mitigate the model degradation with limited success and this strategy still has room for improvement. If we understood correctly, we will try to fit this section in the extra page for the final version.

---

### Official Review · Reviewer_e7ef · 2021-09-21

**Rating:** 6
**Confidence:** 3
**Correctness:** The claims are largely correct, but s…

**Strengths:**

1. This is an insightful paper that investigates several main shortcomings of current approaches.
2. The dataset is creatively constructed and large enough to draw robust conclusions.
3. The experiments are comprehensive and detailed.

**Weaknesses:**

1. How do you ensure that these transformations are actually natural? They seem natural but certain ones occur less likely than others (e.g. Speako does not happen that much in my opinion).
2. How do you choose these transformations? There could be many others that are more realistic but not covered in this benchmark.
3. How well can better ASR systems alleviate these problems? Are the drops in performance due to worse ASR or worse NLP models?

**Additional Feedback:**

none

**Clarity:**

Mostly. Although the part on human evaluation could be more detailed and more clear, I had trouble understanding that.

Several tables like Table 7 could also be improved in presentation, perhaps replaced with graphs/plots to show drops in performance. I'm not quite sure what the main takeaway from Table 7 is.

**Documentation:**

documentation is reasonable

**Relation To Prior Work:**

There can be closer comparison to current work, such as using tables to show comparable dataset sizes, data sources, target studies, and so on.

**Summary And Contributions:**

This paper presents NATURE, a set of simple spoken-language-oriented transformations applied to the evaluation set of datasets, that show the shortcomings of current models in real-world settings. While these transformations are natural human spoken language variations and preserve the semantics of an utterance, they can deteriorate model performance significantly (up to 40%) which shows the limitations of current approaches.

---

> ### Author Response · Authors · 2021-09-29
> **review reply**
>
> In response to 1. how do we ensure the transformations are natural: To ensure that we are as realistic and close to real-case scenarios as possible, we base the transformation's design on academically studied oral phenomena appearing among both native and foreign speakers (see [2], [3], [4], [6], [7], [9], [10]). These recurrent "erratic" phenomena includes the speako (or slip of the tongue) which is not intentional or the product of ignorance and can be quite common and produced by multiple causes (see [8], [9], [10]). From an ASR point of view, it can originate from the environment (e.g., ambient noise) or at the speaker level (e.g., volume, phonetic harmonics, hesitations, repeats, false starts, foreign accents, stutter, lisping and other speech impediments). Surveys and studies in academic research have shown that speakos are part of the speech of both learning speakers (see [5] speakos accounting for 67.4% of oral errors) and native adult speakers (see [5], [11] speakos accounting for 48-54% of oral errors). In another study (see [1]) speakos were seen as such a natural phenomenon that the authors analyzed the effect of modifying a conversational robot's responses across time with speakos to affect human's attitude towards the robot (specially familiarity and sincerity).
> Another way how we ensure our work is (or looks) natural, is by using human evaluation in the form of a survey. It shows that the modified utterances ranked just slightly lower (i.e., 0.5-3.2% for SF) than the control utterances (manually selected to be unambiguous and high quality). The closeness of the score indicating that the original and modified utterances’ labeling is perceived as similar in quality.
>
> In response to 2. how do we choose these transformations: firstly, among all possible transformations, we consider only the least semantically destructive (because we want to maintain the same labels as the original). Then, we keep those that have a broad applicability (are domain-agnostic, can be applied to different kinds of oral-based datasets, can be applied to most types of utterances). Next, we consider the ones that represent a common speech phenomena (as is the case of fillers, synonymy, speakos). Finally, we eliminate all remaining operators that are too complex to implement.
>
> In response to 3. how well can better ASR systems alleviate these problems: we agree that better ASR systems can partially alleviate the considered problems, however not everything can be solved with better ASR. Speech-to-text systems with better noise reduction performance and improved transcription strategies can produce better transcriptions. However, if dataset makers do not acknowledge the difference between spoken and written language and keep “translating” the speech transcriptions to standard written text, the data that is used in research and development to train, tune and evaluate the models will not match the data that the model will encounter in real-life situations, with real users.
>
> Concerning the clarity of table 7: we acknowledge that Table 7 is difficult to grasp at first glance. However, it is the most detailed and compact way we could present the information within the space allotted. It shows in detail the scores obtained on each model (Stack-Prop+BERT and Bi-RNN) and for each dataset variation (Original, Random, Hard) as well as the result obtained when using the same single operator for every utterance. We will do our best to produce a more intelligible and graphical representation of the data in Table 7 and other similar tables for the camera-ready version.
>
> Concerning the relation to prior work stating that there can be closer comparison such as using tables to show comparable dataset sizes, data sources, target studies:  we are not entirely sure to grasp how these comparisons would help understand our research subject. To make a comparison we would need to have enough properties in common to be able to compare them. This is difficult to do when the prior works focus on a whole different task with different sources, forms, objectives, etc. In section 5.2 we do apply to our corpora the code from a prior work that, like us, use alternative evaluation and does not require human-in-the-loop, which would be too costly, and we cover the Data Augmentation approach in our appendix. We have updated the paper to include a small paragraph at the end of the Related Work section highlighting the main similarities and divergences between NATURE and the cited works.
>
> References:
> (see comment below)

---

> > ### Author Response · Authors · 2021-09-29
> > **References**
> >
> > References:
> >
> > [1] A Robot’s Slip of the Tongue: Effect of Speech Error on the Familiarity of a Humanoid Robot (T. Gompei and H. Umemuro, 2015)
> >
> > [2] Exploring filler words and their impact (E. Duvall et al., 2014)
> >
> > [3] Eight main differences between collections of written and spoken data (H.G. TillMann, 1997)
> >
> > [4] Errors and disfluencies in spoken corpora (G. Gilquin and S. De Cock, 2013)
> >
> > [5] Kids' slips: What young children's slips of the tongue reveal about language development (J.J. Jaeger, 2004)
> >
> > [6] On the tip of the tongue: What causes word finding failures in young and older adults? (D.M. Burke et al., 1991)
> >
> > [7] Psychology and language (H.H. Clark and E.V. Clark, 1977)
> >
> > [8] Slip of the tongue in BBC news anchor's videos in textual pronunciation context (D.S. Paradewari and B. Bram, 2020)
> >
> > [9] Slips of the tongue (V.A. Fromkin, 1973)
> >
> > [10] Slips of the tongue : Speech errors in first and second language production (N. Poulisse, 1999)
> >
> > [11] Speech errors in early child language production (J.P. Stemberger, 1989)

---

### Decision · Program_Chairs · 2021-10-09

**Decision:**

Accept

**Comment:**

The paper presents a set of semantics-preserving spoken-language transformations that can be applied to existing datasets. The goal is to provide a mechanism for evaluating the generalizability of learned models in the face of changes to the language that preserve meaning and are consistent with human speech behaviors and ASR-induced errors, in order to measure the overfit of especially deep learning methods on existing benchmarks. Extensive evaluations, comparisons, and analyses are provided, including baselines for other forms of language perturbation. The datasets provided are large enough and well-evaluated enough to be of use, and supports the basic claim that existing models are prone to overfitting on surface patterns. The proposed transformations are intuitive and the paper is well written.